# ADDI: A Simplified E2E Autonomous Driving Model with Distinct Experts and Implicit Interactions

## Abstract

End-to-end autonomous driving has emerged as a promising research trend aimed at achieving autonomy from a human-like driving perspective. Traditional solutions often divide the task into four sub-tasks—tracking-by-detection, online mapping, prediction, and planning—with several interactions to polish planning. However, this modular approach disrupts the cohesion of autonomous driving by decomposing these processes and then linking them through interactions, leading to suboptimal and inefficient practical applications. To address this limitation, we propose ADDI, a simple and efficient end-to-end autonomous driving method. First, ADDI integrates tracking-by-detection and online mapping through a unified detection module paired with distinct expert designs, enabling simultaneous output of detection and mapping elements. Second, ADDI employs a unified motion planning model with distinct experts to jointly predict agent trajectories and ego planning trajectories. With this unified model structure, most interactions required by previous methods are rendered unnecessary. ADDI implements two implicit (resource-free) and two explicit interactions to associate the different components. Experimental results demonstrate that ADDI achieves state-of-the-art performance on both open-loop and closed-loop benchmarks while running significantly faster than prior end-to-end methods.

## 1 Introduction

Autonomous driving is a complex system that requires precise environmental perception and reliable driving behaviors. Traditional methods decompose the autonomous driving system into four sub-tasks—detection, online mapping, prediction, and planning—to accomplish the overall task. However, these methods face challenges related to scalability, cumulative errors, and extensive post-processing. Recently, end-to-end autonomous driving approaches Hu et al. (2023); Jiang et al. (2023); Sun et al. (2024) have introduced a novel framework, enabling autonomous driving to be managed by an end-to-end model. These end-to-end approaches simplify the conventional system, marking the start of a new era in data-driven autonomous driving.

Existing end-to-end methods Hu et al. (2023); Jiang et al. (2023); Jia et al. (2023a); Sun et al. (2024); Jia et al. (2023b) are influenced by traditional autonomous driving pipelines. UniAD Hu et al. (2023) developed end-to-end autonomous driving, it integrates all sub-tasks into a cascade model. Furthermore, VAD Jiang et al. (2023) utilizes a vectorized presentation to eliminate the need for hand-crafted post-processing. Inspired by Lin et al. (2023a), Sun et al. (2024) uses sparse feature extraction to construct 3D features, avoiding the computationally expensive generation of bird's eye view (BEV) features. DriveAdapter Jia et al. (2023a) trains adapters in a frozen reinforcement learning teacher model with imitation learning. Given that planning trajectory is the final target of end-to-end autonomous driving, several methods have improved performance by designing more reasonable planning module strategies Yuan et al. (2023); Song et al. (2024). Prevailing methods typically utilize four tasks. These methods rely on six interactions (Agent-Map, AgentTrajectory-Agent, AgentTrajectory-Map, EgoTrajectory-Agent, EgoTrajectory-Map, AgentTrajectory-EgoTrajectory), and an additional Motion-Temporal interaction to facilitate information exchange between current and previous elements. As shown in fig. 2(Appendix A). They overlook that end-to-end autonomous driving models input comprehensive environmental data, maintaining the integrity of these elements (agents, map elements, agent trajectories, ego planning trajectories, etc.). Disassembling these elements can

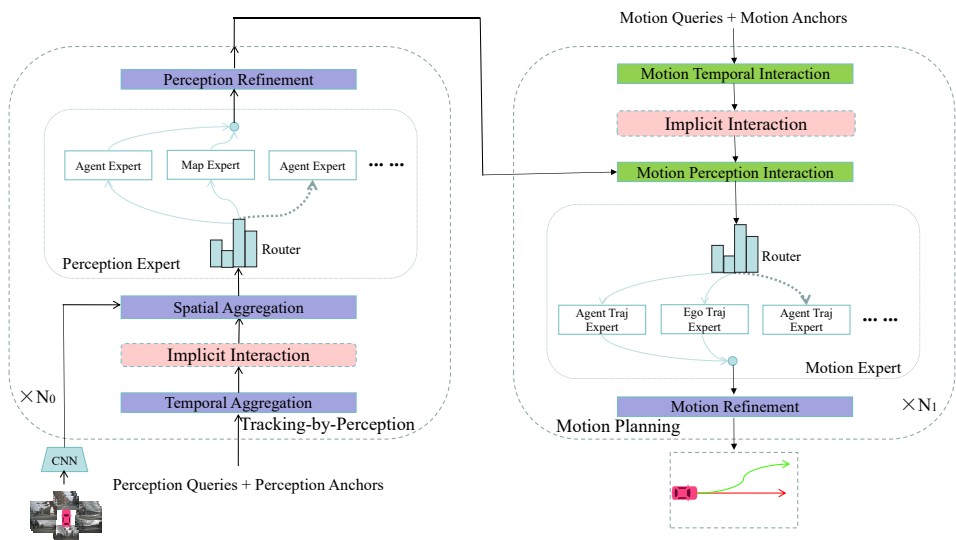

Figure 1: Overview of our newly introduced ADDI. The pipeline of ADDI is composed of Tracking&Perception and Motion Planning. The Tracking&Perception module utilizes a unified model to simultaneously detect agents and map elements. Similarly, the Motion Planning module uses one entity model to export agent predictions and ego motion planning concurrently. ADDI simplifies four parts of traditional end-to-end AD methods into two parts, and utilizes two implicit interactions (without additional resource consumption) and two explicit interactions (Motion Perception Interaction and extra Motion Temporal Interaction) to outweigh the traditional complex interactions.

compromise coherence and disrupt the overall system's integrity. Moreover, the extensive use of sequential models often results in lower inference speeds and higher resource consumption.

In this paper, we propose an efficient and simple end-to-end autonomous driving method, named ADDI, the principle is shown in fig. 2(Appendix A). Since online mapping can be treated as a DETR-like detection method Liao et al. (2023), we manage the online mapping task as tracking&mapping, similar to tracking-by-detection. We combine the detection and mapping tasks with a novel detection module equipped with distinct experts, called Tracking&Perception, which outputs detection and mapping elements simultaneously. Additionally, since both prediction and planning involve trajectory forecasting, we use a unified Motion Planning model with integrated motion experts to predict agent trajectories and ego planning trajectories simultaneously. In summary, our method utilizes only two models to achieve the functionality of four models used in traditional approaches, as illustrated in fig. 1. Our approach also simplifies the complex interactions inherent in traditional end-to-end autonomous driving methods. We replace the six interaction modules and one extra motion-temporal interaction of existing methods with two implicit interactions and two explicit interactions. The two implicit interactions are embedded within the self-attention of Tracking&Perception and Motion Planning, adding no additional resource demands. One explicit interaction is the Motion-Perception Interaction, while the other is a Motion-Temporal interaction that aggregates historical motion information, reflecting the inherent temporal interrelations of vehicle movement. This streamlined design enables our model to accomplish these tasks with two-seventh of the resources, operating more efficiently than previous methods without any performance degradation.

To summarize, the contributions of our paper are as follows:

- We explore task modules for end-to-end autonomous driving and propose an efficient paradigm named ADDI, which unifies multiple tasks with only two modules: Tracking&Perception and Motion Planning. Combined with distinct expert designs, these unified models effectively enhance final results.

- We simplify the complex and cumbersome interaction operations by introducing two implicit interactions and two explicit interactions, replacing traditional interactions and enhancing the temporal feature representation.

- Experiments on the open-loop and closed-loop evaluations demonstrate that our method achieves state-of-the-art performance, showing significant improvements over existing methods and running faster than previous end-to-end approaches.

## 2 RELATED WORK

### 2.1 2D-TO-3D FEATURE TRANSFORMATION

Given surrounding images $P \in \mathbb{R}^{N \times 3 \times H \times W}$, end-to-end autonomous driving systems extract features using specific encoders, which produce features in either Bird's-Eye-View (BEV) or sparse feature form.

#### 2.1.1 BEV ENCODER

BEV-based feature extractors are widely used in autonomous driving Philion & Fidler (2020); Li et al. (2022b); Reiher et al. (2020). Early methods normally convert 3D Euclidean space features to a 2D Euclidean planar surface using the Inverse Perspective Mapping (IPM) Bertozzi et al. (1998); Zhang et al. (2014); Jeong & Kim (2016); Can et al. (2021); Reiher et al. (2020). However, IPM assumes that the height of all objects is zero, which restricts the accurate representation of the surrounding environment. To eliminate the fatal flaw of IPM, LSS Philion & Fidler (2020) implicitly unprojects image features to 3D space with calibrated camera intrinsics and extrinsics. Bev-Former Li et al. (2022b) treats BEV feature extraction as a 3D-to-2D projection task, presenting BEV feature scalars as queries to learn the feature transformations.

#### 2.1.2 SPARSE SAMPLING

Different from dense BEV-based methods, sparse-based algorithms Wang et al. (2021) directly sample sparse features from images. However, their capacity is limited; for example, DETR3D Wang et al. (2021) samples features from single 3D reference points, which restricts its ability to learn representations from a global perspective. Similarly, PETR Liu et al. (2022) transfers the 2D features into the 3D representation by encoding 3D position embedding without complex 2D-to-3D projection and feature sampling. In addition, Sparse4D Lin et al. (2023a) utilizes multiple sparse keypoints distributed across 3D anchor box regions for feature sampling.

### 2.2 MULTI-TASK DECODER

End-to-end autonomous driving task consists of multiple decoders.

#### 2.2.1 DETECTION DECODER

3D object detection is a foundational task in end-to-end autonomous driving. Early methods Wang et al. (2021); Liu et al. (2020); Duan et al. (2019) typically detect 3D objects from a single image. Recently, multi-view 3D detection has gained popularity and significantly advanced the field of perception. Philion & Fidler (2020); Li et al. (2022b); Huang & Huang (2022) detect objects from BEV features with anchor-free strategies or DETR-like detection heads Yin et al. (2021); Tian et al. (2019); Wang et al. (2021). Liu et al. (2022); Wang et al. (2023); Lin et al. (2023a) utilize 3D positional encodings and detection queries to learn object features via attention mechanisms.

#### 2.2.2 ONLINE MAPPING

Online mapping provides static environmental perception. HDMapNet Li et al. (2022a) predicts semantic segmentation results on BEV features; however, it requires complex post-processing to generate vectorized HD maps. VectorMapNet Liu et al. (2023) was the first method to predict sequential sampling points of map elements. Subsequently, Liao et al. (2023); Liu et al. (2024); Zhou et al. (2024); Zhang et al. (2024); Hu et al. (2024) utilize DETR-like transformer structures that sample map elements as point sets using a group of fixed permutations, then the hierarchical queries are responsible for extracting structured map elements. Additionally, Yuan et al. (2023); Chen et al. (2024a) associate tracked HD map elements from historical frames to achieve more accurate map constructions.

### 2.2.3 TRACKING

Recently, transformer-based attention mechanisms with track queries are widely adopted in end-to-end multi-object tracking to associate instances across frames Meinhardt et al. (2022); Sun et al. (2021); Cai et al. (2022); Gao & Wang (2024). These methods achieve significant performance improvements in tracking tasks. Inspired by these methods, Lin et al. (2023b); Sun et al. (2024) extend tracking queries by incorporating memory mechanisms to enhance performance further.

### 2.2.4 PREDICTION

As a crucial part of autonomous driving, motion prediction is used to forecast the future trajectories of detected objects. Early works predict future trajectories with simple neural networks Luo et al. (2020); Casas et al. (2021a). in addition, Liu et al. (2021); Gao et al. (2020) introduce sparse representations to predict the dynamic object behavior. Recently, PnPNet Liang et al. (2020) introduced a novel learnable process to address both data association and trajectory estimation. Gu et al. (2023); Jiang et al. (2022) predict future trajectories by interacting with tracked objects and vectorized maps, showing significant improvements in experimental results.

### 2.2.5 PLANNING

With the support from previous tasks, the ultimate goal of end-to-end methods is to predict planning trajectories or control signals. Early works predict trajectories directly without interacting with perception and motion prediction Pomerleau (1988); Prakash et al. (2021), resulting in unsatisfactory performance. Subsequently, Casas et al. (2021b); Cui et al. (2021); Sadat et al. (2020); Hu et al. (2022) achieve significant progress by utilizing dense cost maps constructed from perception and motion predictions. However, these methods required hand-crafted rules to select the optimal trajectory based on cost maps. Additionally, reinforcement learning is used to predict planning trajectories Toromanoff et al. (2020); Chekroun et al. (2022); Chen et al. (2021). Notably, recent research introduces unified frameworks that integrate perception, prediction, and planning Hu et al. (2023). Furthermore, Jiang et al. (2023); Ye et al. (2023) predict agent motion by interacting with dynamic tracking objects and static map elements, refining the planning with several constraints. In contrast to prior methods, SparseDrive Sun et al. (2024) constructs a multi-task with sparse feature sampling, it also utilizes a hierarchical planning selection strategy that incorporates a collision-aware rescore module to obtain a rationality and safety planning result. Recent studies have noted that open-loop data cannot adequately simulate vehicle interactions in autonomous driving contexts. Consequently, several methods use closed-loop datasets to evaluate and optimize each module, assessing performance based on Driving Score, Route Completion, and Infraction Score Li et al. (2024); Jia et al. (2023a;b); Chen et al. (2024b).

## 3 METHOD

### 3.1 OVERVIEW

The end-to-end autonomous driving framework of ADDI is depicted in fig. 1. Our DETR-like network is simplified with three components: Tracking&Perception, Motion Planning, and Constraint Loss. Specifically, given a series of surrounding images, our Tracking&Perception module, abbreviated as the perception module, extracts features from these images and projects them into a 3D sparse space to represent perception queries (for dynamic agents and static map elements). These queries are then fed into our perception expert module to enforce the feature representation, followed by a refinement module for further processing. Next, these perception queries are passed to the motion planning module, which generates motion trajectories (for both agents and the ego vehicle) by interacting with historical motion features and perception outputs. In addition, the motion planning module uses a motion expert design similar to the perception expert, to finetune the ego planning trajectories and agent predictions. Finally, we use the constraint loss to optimize the overall network.

Notably, our method utilizes two implicit interactions (self-attention in Tracking&Perception and Motion Planning) and two explicit interactions (Motion-Temporal Interaction, Motion-Perception Interaction) to outweigh the six traditional interactions (Agent-Map, AgentTrajectory-Agent, AgentTrajectory-Map, EgoTrajectory-Agent, EgoTrajectory-Map, AgentTrajectory-EgoTrajectory)

and the extra Motion-Temporal interaction of existing methods, achieving a more streamlined and efficient approach.

## 3.2 TRACKING&PERCEPTION

Prior works Hu et al. (2023); Jiang et al. (2023); Sun et al. (2024), normally divide the perception task into two sub-tasks: tracking-by-detection and mapping. In our perception task, we propose tracking&perception with two improvements: we conduct the map construction to tracking tasks, similar to tracking-by-detection; we use a single module to simultaneously predict dynamic objects and static map elements.

### 3.2.1 PERCEPTION QUERIES INITIALIZATION

We first initialize the perception queries and perception anchors based on their unique characteristics. Specifically, perception anchors are composed of agent anchors and map element anchors. Agent anchors (N x 11) are formatted with position, dimension, direction, and velocity: $x, y, z, w, h, l, sin(yaw), cos(yaw), v_x, v_y, v_z$. Map element anchors (N x 40) are composed by N sampled vector points: $x_1, y_1, x_2, y_2, \ldots, x_{20}, y_{20}$. These anchors are then added to perception queries, with each anchor encoded through a unique embedding layer. The simplified expression of this process is given by:

$$Q_u = Q_p + concat(EB_a(Anchor_a), EB_m(Anchor_m)) \tag{1}$$

where $Q_u$ denotes the unified perception queries, $Q_p$ is the initialized query features, $EB_a$ and $EB_m$ are the agent embedding encoder and map element embedding encoder respectively, $Anchor_a$ is the agent anchors, $Anchor_m$ is the map element anchors.

### 3.2.2 PERCEPTION MODEL UNIFICATION

After extracting image features using ResNet and FPN He et al. (2015); Lin et al. (2017), we designed our perception module with four components: Temporal Aggregation, Spatial Aggregation with Implicit Interaction, Perception Expert, and Perception Refinement.

**Temporal Aggregation.** Historical features play a crucial role in understanding temporal scenes. Therefore, we design a perception memory bank to cache historical features. Our perception memory bank is based on the memory bank design from Lin et al. (2023b), differently, we cache both historical agent features and map elements features. In addition, we use cross-attention to aggregate the historical information into the current frame.

$$Q_{ta} = cross\text{-}atten(Q_u, Q_t) \tag{2}$$

where, $Q_u$ is the unified perception queries, $Q_t$ denotes the historical queries, and $Q_{ta}$ presents the temporally aggregated output.

**Spatial Aggregation with Implicit Interaction.** Spatial Aggregation is composed of self-attention and cross-attention. Since unified perception queries consist of agent queries and map element queries, self-attention implicitly captures the interaction between agents and map elements, allowing it to replace traditional agent-map interactions. This design not only provides a more coherent representation of agent-map relationships but also conserves resources. Cross-attention is used to extract spatial features from regions of interest across image features using intrinsics and extrinsics. The process is described as follows:

$$Q_{sa} = cross\text{-}atten(self\text{-}atten(Q_{ta}), F_{img}) \tag{3}$$

where, $Q_{ta}$ is the previous temporally aggregated output, $F_{img}$ denotes the image features, and $Q_{sa}$ is the spatially aggregated output.

**Perception Expert.** Since we use a unified model to predict both agents and map elements simultaneously, using an identical approach to fit both agents and map elements with significantly different structural dimensions, is inherently imprecise. Inspired by Mixture of Experts Shazeer et al. (2017), which selects different parameters for each incoming example, we have designed perception experts

to separately enforce agent and map element features. The overview of the perception expert is illustrated on the left side of fig. 1. This design is composed of rooters and agent-map experts. Concretely, our router primarily directs the representation from the previous module to the best-determined top-K experts selected from a set $\{E_i(x)\}_{i=0}^{N-1}$, where $N$ is the total number of experts, and $E_i$ is the $i$-th expert. Typically, we skip computing outputs from experts with zero routes, which reduces computational costs. The router operation normalizes the selection via a softmax distribution over the top-K logits, as illustrated below:

$$y = \sum_{i=0}^{N-1} SoftMax(TopK(W_r \cdot x))_i \cdot E_i(x) \tag{4}$$

Where $x$ is the perception queries from the previous module. $E_i(x)$ is the $i-th$ sparse map element expert. The router variable $W_r$ produces logits via $W_r \cdot x$. The value $K$ of top-K is a hyper-parameter that modulates the number of experts used for process perception queries. This design has a notable success in computational efficiency, which means that even if we increase $N$ while keeping $K$ fixed, the model's parameter count increases while the computational cost remains constant. This motivates a distinction between the model's total parameter count and the number of parameters used for processing an individual active parameter count. As shown in fig. 1, we design two types of experts: the agent expert and the map element expert. The outputs from these perception experts enable simultaneous regression of dynamic agents and map elements.

**Perception Refinement.** Finally, we utilize a perception refinement module to further refine the predictions of agent anchors and map element anchors. Selected features are also stored in a perception memory bank. This refinement module is conducted with a convolution and batch-norm layer ($Conv\text{-}Bn$), followed by the addition of the perception anchors, as illustrated below:

$$P_o = Conv\text{-}Bn(y_p) + Anchor_{a+m} \tag{5}$$

where $P_o$ is the refinement output, $y_p$ is the output of last module, $Anchor_{a+m}$ is the anchor of perception.

## 3.3 MOTION PLANNING

Existing methods often address motion prediction and motion planning independently, treating each task as a standalone process, overlooking the similarity of these two tasks, both of which involve future trajectory prediction, and leading to inefficient use of computational resources. In this work, we use a unified motion trajectory model to simultaneously predict the agent trajectories and ego planning trajectories. Since we use the unified model to predict agent trajectories and ego planning trajectories, we introduce unified queries, named motion queries, to integrate agent trajectory queries and ego planning queries.

### 3.3.1 MOTION QUERIES INITIALIZATION

In our unified motion model, different queries must be harmonized. Our Motion queries are composed of agent trajectory queries and ego planning queries. Specifically, agent trajectory queries inherit the perception agent queries with the addition of anchor embeddings. Ego planning queries are different, we encode surrounding image features using a convolutional layer and incorporate features extracted from historical ego trajectories. Given the fixed dimensions, position, and vehicle dynamics of the ego vehicle, the ego anchor is embedded through an embedding layer and added to the ego queries. This process can be formulated as:

$$Q_p = concat(Q_a, Q_e + embedding_e(anchor_e)) \tag{6}$$

Here, $Q_p$ denotes the motion queries, $Q_a$ is the agent trajectory queries inherited from the perception agent features, $Q_e$ is the ego query extracted from surrounding image features and history trajectories, $anchor_e$ is the ego vehicle information embedded with anchor encoder $embedding_e$.

### 3.3.2 PLANNING MODEL UNIFICATION

As depicted on the right of fig. 1, our motion model is composed of multiple stacked motion layers. Each motion layer includes the Motion-Temporal Interaction, Motion-Perception Interaction with

Implicit Interaction, Motion Expert, and Refinement, collectively performing temporal modeling, interaction, and result refinement.

**Motion-Temporal Interaction.** The Motion-Temporal Interaction is designed to progressively integrate temporal information to improve the trajectory continuity and smoothness. This integration is essential, as historical information significantly influences the understanding of temporal motion behaviors of motion queries. We use a memory bank to store two types of information: historical motion query features and historical motion trajectories. The historical motion query features gather information about the historical surrounding environment, thereby reinforcing diverse scene representations. The historical motion trajectories serve as the successful routes to improve the accuracy of current trajectory predictions. Our memory bank maintains a set of $n$ historical records projected onto current coordinates. Unlike Li et al. (2022b); Chen et al. (2024a), which directly stacks the historical features and current features, we aggregate historical information with a cross-attention transformer, which extracts and integrates core information for current features. This procedure is defined as follows:

$$F_{mt} = cross\text{-}atten(F_c, CM(F_h, T_h)) \tag{7}$$

where $F_{mt}$ is the output of Motion-Temporal Interaction, $cross\text{-}atten$ refers to cross attention transformer, $CM$ is the confidence mask, $F_c$ denotes the current motion query features, $F_h$ and $T_h$ are historical motion query features and historical trajectories respectively.

**Motion-Perception Interaction with Implicit Interaction**

Similar to our Tracking&Perception module, our motion planning module integrates agent trajectory queries with ego planning queries, implicitly modeling the interaction between agent trajectories and the ego trajectory using a self-attention mechanism. Our Motion-Perception Interaction enables us to streamline conventional operations, such as Agent-Map, AgentTrajectory-Agent, AgentTrajectory-Map, EgoTrajectory-Agent, EgoTrajectory-Map, and AgentTrajectory-EgoTrajectory interaction with one interaction module. In effect, this design allows us to accomplish these tasks with one-sixth of the resources. Our Motion-Perception Interaction progressively extracts information from the perception module, thereby enhancing motion trajectories. This interaction is implemented through a cross-attention transformer module, the process is formulated as

$$F_{mp} = cross\text{-}atten(F_{mt} + AE_{mt}, F_p + AE_p) \tag{8}$$

where $F_{mp}$ presents the output of Motion-Perception Interaction, $cross\text{-}atten$ is the multi-head cross attention, $F_{mt}$ is the result from previous module Motion-Temporal Interaction, $F_p$ denotes the feature of perception module, $AE_{mt}$ and $AE_p$ are Motion-Perception Interaction anchor embeddings and perception embeddings respectively.

**Motion Expert.** To distinguish between agent trajectories and the ego trajectory, we use a specialized expert module similar to the Perception Expert module, but with distinct hyper-parameters. We design two types of experts: the agent trajectory expert and the ego trajectory expert.

**Motion Refinement.** Finally, a motion trajectory refinement module is employed to fine-tune trajectory predictions. Concretely, we use a decoupled MLP head to generate offsets and update initial predictions. The procedure is defined as follows:

$$T_a, T_e = DH(Q_t + RE_t) \tag{9}$$

Here, $T_a$ and $T_e$ denote the predicted agent trajectory and ego trajectory planning, $DH$ is our decoupled MLP head, $Q_t$ presents the output of previous Motion Expert module, $RE_t$ is the refinement embedding.

## 4 EXPERIMENTS

### 4.1 DATASETS

We evaluate our ADDI on the nuScenes Caesar et al. (2020) for open-loop evaluation and the CARLA dataset adopted from the CARLA simulator Dosovitskiy et al. (2017) for closed-loop evaluation. The nuScenes dataset is a real-world autonomous driving dataset consisting of 1000 scenes, annotated at 2 Hz. Each frame collected data from six synchronized surrounding cameras, a Li-DAR, etc. The CARLA dataset is based on CARLA Leaderboard v2. To reduce the differences

Table 1: **Open-loop evaluation of planning performance on the nuScenes dataset.** ADDI notably outperforms other methods across evaluation metrics. Furthermore, ADDI achieves the efficient inference speed compared with other methods. The best results of our method are highlighted in bold. LiDAR-based methods are indicated with †. *: Reproduced with official checkpoint. The inference speed is measured with NVIDIA Tesla A800 GPU.

| Methods | L2 (m) ↓ | | | | Collision Rate (%) ↓ | | | | FPS ↑ |
|---|---|---|---|---|---|---|---|---|---|
| | 1s | 2s | 3s | Avg. | 1s | 2s | 3s | Avg. | |
| FF† Hu et al. (2021) | 0.55 | 1.20 | 2.54 | 1.43 | 0.06 | 0.17 | 1.07 | 0.43 | – |
| EO† Khurana et al. (2022) | 0.67 | 1.36 | 2.78 | 1.60 | 0.04 | 0.09 | 0.88 | 0.33 | – |
| ST-P3Hu et al. (2022) | 1.33 | 2.11 | 2.90 | 2.11 | 0.23 | 0.62 | 1.27 | 0.71 | 1.6 |
| UniADHu et al. (2023) | 0.48 | 0.96 | 1.65 | 1.03 | 0.05 | 0.17 | 0.71 | 0.31 | 1.8 |
| VADJiang et al. (2023) | 0.41 | 0.70 | 1.05 | 0.72 | 0.07 | 0.17 | 0.41 | 0.22 | 4.5 |
| UniAD* | 0.45 | 0.70 | 1.04 | 0.73 | 0.62 | 0.58 | 0.63 | 0.61 | 1.8 |
| VAD* | 0.41 | 0.70 | 1.05 | 0.72 | 0.07 | 0.17 | 0.41 | 0.22 | 4.5 |
| AD-MLP Zhai et al. (2023) | 0.53 | 0.91 | 1.48 | 0.97 | 0.17 | 0.46 | 0.83 | 0.49 | – |
| FusionADYe et al. (2023) | – | – | – | – | 0.02 | 0.08 | 0.27 | 0.12 | 1.6 |
| PPADChen et al. (2024c) | 0.30 | 0.69 | 1.26 | 0.75 | 0.03 | 0.22 | 0.73 | 0.33 | 2.6 |
| ParaDrive Weng et al. (2024) | 0.26 | 0.59 | 1.12 | 0.66 | 0.00 | 0.12 | 0.65 | 0.26 | – |
| SparseDriveSun et al. (2024) | 0.29 | 0.63 | 0.97 | 0.63 | 0.03 | 0.09 | 0.19 | 0.10 | 5.5 |
| Proposed-small | 0.30 | 0.60 | 0.98 | 0.63 | 0.03 | 0.10 | 0.20 | 0.11 | 7.4 |
| Proposed | 0.29 | **0.53** | **0.90** | **0.57** | 0.02 | **0.06** | **0.14** | **0.07** | **6.2** |

Table 2: **Closed-loop evaluation of planning performance on CARLA dataset.** Comparison with existing methods. ADDI notably outperforms other methods across all metrics. The best results of our method are highlighted in bold. C presents the camera and L denotes the LiDAR. All experiments are reproduced with our collected CARLA dataset. DS: Driving Score, RC: Route Completion, IS: Infraction Score.

| Methods | Modality | DS ↑ | RC ↑ | IS ↑ |
|---|---|---|---|---|
| UniADHu et al. (2023) | C | 36.89 | 51.07 | 0.73 |
| VADJiang et al. (2023) | C | 48.26 | 63.20 | 0.78 |
| ThinkTwiceJia et al. (2023b) | C+L | 52.67 | 69.33 | 0.75 |
| DriveAdapter+TCPJia et al. (2023a) | C+L | 58.44 | 74.64 | 0.79 |
| SparseDriveSun et al. (2024) | C | 59.31 | 74.38 | 0.80 |
| Proposed | C | **64.52** | **78.10** | **0.82** |

from nuScenes datasets, we configured sensors similarly to those used in nuScenes (6 surrounding cameras, 1 LiDAR, 1 GNSS, and HD-Maps). To tackle the problem of multi-vehicle interaction scenarios, we use Jia et al. (2024) and Krajzewicz et al. (2012) to simulate complex urban transportation scenarios, including scenarios with sudden lane changes, abrupt braking, turns, and traffic accidents. This strategy could make our simulation scenario similar to the real scenario. We collected the CARLA dataset from 15 towns, each containing multiple scenarios, with each scenario comprising several clips similar to those in nuScenes.

Table 3: Ablation studies were conducted on the key design elements of ADDI using the CARLA dataset. The results demonstrate that each modification contributes to an improvement in performance. All experiments are reproduced with our collected CARLA dataset.

| ID | + Implicit | + P-M Explicit | + M-T Explicit | + P Expert | + M Expert | DS ↑ | RC ↑ | IS ↑ |
|---|---|---|---|---|---|---|---|---|
| 1 | ✔ | | | | | 40.50 | 57.63 | 0.71 |
| 2 | ✔ | ✔ | | | | 52.07 | 67.56 | 0.75 |
| 3 | ✔ | ✔ | ✔ | | | 59.82 | 76.93 | 0.80 |
| 4 | ✔ | ✔ | ✔ | ✔ | | 62.83 | 77.32 | 0.82 |
| 5 | ✔ | ✔ | ✔ | ✔ | ✔ | **64.52** | **78.10** | 0.82 |

## 4.2 COMPARISONS WITH STATE-OF-THE-ART METHODS

We implemented our method based on Jiang et al. (2023); Jia et al. (2024); Sun et al. (2024) and evaluated it against state-of-the-art methods in both open-loop and closed-loop. table 1 and table 12 show the open-loop experiment results, table 2 illustrate the closed-loop experiment results. Furthermore, we tested the inference speed of ADDI and comparison methods using a single A800 GPU.

### 4.2.1 PERFORMANCE ON OPEN-LOOP

To ensure a comprehensive analysis, we compared ADDI's performance with existing methods. As illustrated in table 1, our method achieves significant advantages in both performance and speed on the nuScenes dataset. Concretely, ADDI reduces the average planning L2 distance error by 0.06m compared to Sun et al. (2024) and outperforms competing methods by over 30% in average collision rates. Our model's unified design and implicit and explicit interaction features enable the fastest inference speed among tested methods, achieving 1.13× faster processing while maintaining superior planning performance. The ADDI-small variant runs 1.35× faster than the leading method Sun et al. (2024) while maintaining a comparable performance.

### 4.2.2 PERFORMANCE ON CLOSED-LOOP

In the CARLA validation benchmarks for closed-loop evaluation, ADDI surpassed state-of-the-art methods, as illustrated in table 2, ADDI achieves a Drive Score of 64.52, Route Completion of 78.10, and Infraction Score of 0.82, significantly improving by 5.21, 3.46, and 0.02 respectively, relative to existing best methods. Differing from the nuScenes scenarios, which primarily feature straight lanes and simple interactions, CARLA dataset scenarios are more complex, this closed-loop experiment proved that ADDI achieves superior performance in complex scenarios while maintaining shorter inference times.

## 4.3 ABLATION STUDY

### 4.3.1 KEY COMPONENTS DESIGN

We conducted several ablation studies on the closed-loop CARLA dataset to assess the effectiveness of the proposed modules. We only cascade the Tracking&Perception and Motion Planning module without expert components as our baseline, as illustrated in ID-1 of table 3, then adding key interaction components and expert components progressively, the influence of each component is presented in table 3. Notably, the baseline (ID-1) indicates that all experiments integrate the implicit interaction, the reason is that the implicit interaction is concealed in the self-attention of our unified modules, which is non-removable. The second row (ID-2) shows that Perception-Motion Interaction induces Driving Score and Route Completion by approximately 28.6% and 17.2% over the baseline. As shown in ID-3, the exhaustive utilization of Motion-Temporal Interaction results in a noticeable improvement, yielding 7.75, 9.37, and 0.05 in terms of Driving Score, Route Completion, and Infraction Score. Adding the Perception Expert further enhances feature representation, resulting in a 3.01 increase in Driving Score and 0.39 in Route Completion. Similarly, the last variant shows that Motion Expert also boosts performance.

We performed additional quantitative and qualitative experiments, for comprehensive details, please refer to **Appendix A and B**.

## 5 CONCLUSION

As an efficient end-to-end autonomous driving method, ADDI simplifies the traditional four tasks (tracking-by-detection, online mapping, prediction, and planning) into two stages (Tracking&Perception and Motion Planning). ADDI also utilizes two implicit interactions and two explicit interactions, reducing the complexity of existing interactions. Since implicit interactions are concealed in the self-attention of this model, they incur no additional computational cost. Extensive experiments demonstrate that our method significantly outperforms existing methods in both inference speed and accuracy.

## 6 REPRODUCIBILITY STATEMENT

our method is reproducible, we provide comprehensive instructions and code, reproducing large-scale experiments. All datasets used are publicly available, and we provide preprocessing scripts where necessary. Hyperparameters, training details, and evaluation protocols are described in the paper and included in the code repository. Our experiments can be reproduced on a 8 GPU within the reported computational budget.

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

# A   APPENDIX A

## A.1   COMPARISON OF EXISTING METHODS WITH ADDI

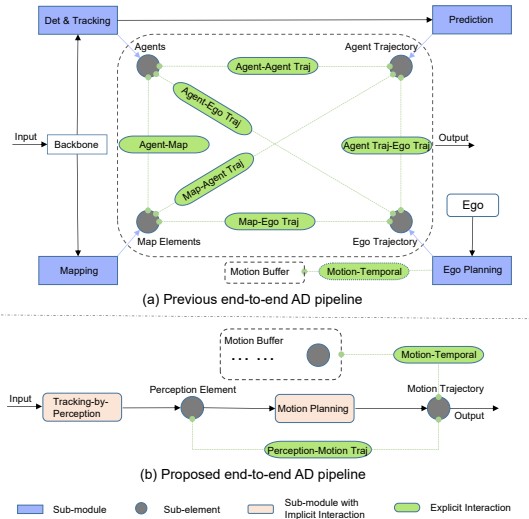

Figure 2: Comparison of existing methods with ADDI. (a) Previous popular end-to-end AD pipeline, like UniAD, VAD, SparseDrive. (b) Proposed end-to-end AD pipeline. Our method uses two modules—tracking&perception and motion planning with implicit interactions—to replace the original four modules and utilizes a single explicit interaction instead of the six interactions previously required. We also introduce a motion-temporal interaction to aggregate historical features. This design enables our model to achieve efficient and competitive performance.

## A.2   LOSS

We use a base loss function similar to Sun et al. (2024); Jiang et al. (2023), adding an extra two expert balance losses as our training loss. The total loss is defined as follows:

$$l_{total} = l_{det} + l_{map} + l_{mot} + l_{pred} + l_{plan} + l_{bp} + l_{bm} \tag{10}$$

where $l_{det}$ denote the detection loss, $l_{map}$ is the map construction loss, $l_{mot}$ is the tracking loss, $l_{pred}$ and $l_{plan}$ present the agent trajectory prediction loss and ego planning loss, $l_{bp}$ and $l_{bm}$ are perception expert balance loss and motion expert balance loss.

The expert balance loss $l_{bp}$ and $l_{bm}$ are identical and encourage balanced contributions from each expert, preventing scenarios where one expert processes all inputs while others are rarely engaged. We ensure an even distribution of workload with a simplified formulation as follows:

$$l_b = \alpha \cdot N \cdot \sum_{i=0}^{N-1} f_i \cdot P_i \tag{11}$$

Here, $N$ denotes the number of experts indexed by $i = 0$ to $N - 1$, and $T$ is the number of tokens. The variable $f_i$ represents the percentage of inputs routed to each expert, $\alpha$ is a hyper-parameter, and $P_i$ denotes the fraction of routing probability assigned to each expert.

### A.3 INTERMEDIATE SUB-TASK EVALUATION

To ensure a fair comparison with state-of-the-art methods, we conducted sub-task experiments on the nuScenes validation set. The evaluation metrics for each task are detailed below.

**Detection.** The agent detection results from the Tracking&Perception module are presented in table 4. ADDI achieves 0.512 mAP and 0.593 NDS, surpassing SparseDrive Sun et al. (2024) by 3% and 1%, respectively.

**Multi-object Tracking.** Building on the excellent detection results, the tracking evaluation yields 0.533 AMOTA, 1.052 AMOTP, and 616 IDS, which surpass the existing method by 6%, 3%, 3% in terms of AMOTA, AMOTP, and IDS. The results are reported in table 5.

**Online mapping.** As shown in table 11, ADDI delivers expected performance, achieving a 1.5% improvement in pedestrian Average Precision ($APped$) compared to SparseDrive Sun et al. (2024). However, a slight decrease in overall mAP is observed due to the unified model's equal emphasis on agents and map elements.

**Prediction.** For agent prediction derived from motion planning results, table 7 illustrates that our model outperforms the existing method by 3.3% and 3.4% in minADE and EPA respectively.

In summary, ADDI enforces the implicit representation of various elements, such as agents, map elements, agent trajectories, and ego-planning trajectories. This interconnected framework enhances final planning decisions while maintaining competitive performance across intermediate sub-task evaluations.

Table 4: **Detection results on the nuScenes dataset.** ADDI achieves the best perfomance on detection tasks among end-to-end methods.

| Methods | mAP ↑ | mATE ↓ | mASE ↓ | mAOE ↓ | mAVE ↓ | mAAE ↓ | NDS ↑ |
|---|---|---|---|---|---|---|---|
| UniADHu et al. (2023) | 0.380 | 0.684 | 0.277 | 0.383 | 0.381 | 0.192 | 0.498 |
| SparseDriveSun et al. (2024) | 0.496 | 0.543 | 0.269 | 0.376 | 0.229 | 0.179 | 0.588 |
| Proposed | **0.512** | **0.537** | 0.276 | **0.367** | **0.210** | **0.179** | **0.593** |

Table 5: **Multi-object tracking.** Following the evaluation criteria used in Hu et al. (2023)

| Methods | AMOTA ↑ | AMOTP ↓ | Recall ↑ | IDS ↓ |
|---|---|---|---|---|
| UniADHu et al. (2023) | 0.359 | 1.320 | 0.467 | 906 |
| SparseDriveSun et al. (2024) | 0.501 | 1.085 | 0.601 | 632 |
| Proposed | **0.533** | **1.052** | **0.628** | **616** |

Table 6: **Online mapping with tracking strategy.** Comparison with state-of-the-art method on the nuScenes dataset.

| Methods | $AP_{ped}$ ↑ | $AP_{div}$ ↑ | $AP_{bound}$ ↑ | mAP ↑ |
|---|---|---|---|---|
| VADJiang et al. (2023) | 40.6 | 51.5 | 50.6 | 47.6 |
| SparseDriveSun et al. (2024) | 53.2 | 56.3 | 59.1 | 56.2 |
| Proposed | **54.0** | 55.7 | 57.9 | 55.9 |

Table 7: **Prediction.** Split the agent predictions from our motion planning module, and compare with state-of-the-art methods.

| Methods | minADE(m)↓ | minFDE(m)↓ | MR↓ | EPA↑ |
|---|---|---|---|---|
| UniADHu et al. (2023) | 0.71 | 1.02 | 0.151 | 0.456 |
| SparseDriveSun et al. (2024) | 0.60 | 0.96 | 0.132 | 0.555 |
| Proposed | **0.58** | 0.97 | **0.130** | **0.574** |

## A.4 EXPERT QUOTA.

To further investigate the impact of our expert, we performed studies focusing on the expert quota. As shown in Figure 3, we evaluated models with varying numbers of experts per layer. Ablation studies were performed using a brief training duration of 35 epochs. We considered 12 models with identical depths, with the number of experts per layer set to 3, 4, 8, and 16. The results showed that the performance of both the perception and motion experts peaked with 8 experts per layer and 2 routers. The blue line is the result of using FFN in both perception and motion planning modules. Notably, the 2/4 and 2/8 expert configurations outperformed FFNs on the nuScenes dataset (using twice the computing resources and time).

## A.5 MAP ANCHOR QUOTA

In this study, we vary the number of initialized and cached map anchors. As shown in the table, our method achieves peak performance on the nuScenes dataset when using 50 initialized map anchors and 100 cached anchors. This can be attributed to the fact that, in nuScenes scenarios, the number of map elements typically does not exceed 50. Using more than 50 initialized map anchors may introduce noise, thereby degrading model performance.

## A.6 OPEN-LOOP EVALUATION ON CARLA

Since the nuScenes dataset primarily contains simple driving scenarios, we further evaluated our method on the CARLA dataset, adopted from the CARLA simulator Dosovitskiy et al. (2017), which simulates complex urban transportation scenarios, including vehicles suddenly inserting, braking, turning, and traffic accidents. As shown in table 12, the evaluation metrics results are lower than the results in table 1. This discrepancy arises from the more complex and varied scenarios in CARLA, closely resembling real urban environments. Nonetheless, our method outperforms the best existing method Hu et al. (2023), achieving a 4.1% improvement in L2 distance and an 8.7% reduction in collision rate.

## A.7 EFFICIENCY

To analyze the inference efficiency of ADDI, we experimented, as shown in table 8. Compared to existing state-of-the-art methods, ADDI delivers superior performance with fewer parameters using the same backbone. Achieving a processing speed of 6.2 FPS, ADDI is faster and more efficient than competing methods.

Table 8: **Efficiency comparisons.** ADDI notably outperforms other methods in terms of parameters and FPS. The best results of our method are highlighted in bold. Experiments are measured on 1 NVIDIA Tesla A800 GPU.

| Methods | FLOPs (G) | Params (M) | FPS |
|---|---|---|---|
| UniADHu et al. (2023) | 1709 | 125 | 1.8 |
| VADJiang et al. (2023) | – | – | 4.5 |
| SparseDriveSun et al. (2024) | 790 | 105 | 5.5 |
| Proposed | **763** | **102** | **6.2** |

Table 9: **Effectiveness of Model Unification: Experiments on the closed-loop CARLA dataset.** In this experiment, our baseline configuration involves disassembling the two unified modules in ADDI (perception unification and planning unification) and replacing them with four traditional, separate modules (detection, mapping, prediction, and planning). "UP" represents unifying the detection and mapping with our Tracking&Perception module. "UM" presents unifying the prediction and planning with our Motion Planning module. During this experiment, the explicit interactions are similar to ADDI.

| ID | UP | UM | Driving Score ↑ | Route Completion ↑ | Infraction Score ↑ |
|---|---|---|---|---|---|
| 1 | | | 36.03 | 52.45 | 0.70 |
| 2 | ✔ | | 52.19 | 67.33 | 0.75 |
| 3 | ✔ | ✔ | **64.52** | **78.10** | **0.82** |

## A.8 EFFECTIVENESS OF MODEL UNIFICATION

As shown in table 9, we conducted experiments on the closed-loop CARLA dataset to assess the effectiveness of model unification. We replaced the two unified modules in ADDI (perception unification and planning unification) with the traditional four separate modules (detection, mapping, prediction, and planning) as the baseline (ID-1), while preserving similar explicit interactions to ADDI. The second row (ID-2) shows that the unified perception module (Tracking&Perception) increases Driving Score and Route Completion by approximately 45% and 28% over the baseline. As shown in ID-3, incorporating the unified planning module (Motion Planning) further enhances feature representation, resulting in a 12.33 increase in Driving Score and a 10.77 improvement in Route Completion.

## A.9 EFFECTIVENESS OF REFINEMENT

We also trained for 35 epochs to verify these two refinements (Perception Refinement and Motion Refinement) against the baseline (without the convolution operation, only with the concatenate operation), showing a slight improvement in performance, as shown in table 10.

# B APPENDIX B

## B.1 QUALITATIVE RESULTS

We present additional qualitative results to substantiate the superior performance of our model further. Qualitative results from the CARLA closed-loop experiments are presented in fig. 4, fig. 5 and fig. 6, with the left side displaying six surrounding images and the right side showing the top-down view of ego vehicles. Our method predicts more accurate planning trajectories overall, with fewer false negatives and false positives, especially in challenging scenarios such as avoiding pedestrians, extreme weather, and turnings.

Table 10: **Effectiveness of Refinement: Experiments on the closed-loop CARLA dataset.** Our baseline ID-1 configuration replaces our two refinement modules in ADDI (perception refinement and trajectory refinement) with the non-convolutional operation (using a simple add operation). "PR" represents perception refinement. "TR" presents trajectory refinement.

| ID | PR | TR | Driving Score ↑ |
|----|----|----|----|
| 1 |  |  | 29.17 |
| 2 | ✔ |  | 30.85 |
| 3 | ✔ | ✔ | 33.33 |

Table 11: **Studies on the map anchor quota.**

| Init Num/Cached Num | Cached Num—L2(m)avg ↓ | Collision Rate avg ↓ |
|----|----|----|
| 50/50 | 0.68 | 0.11 |
| 50/100 | 0.57 | 0.07 |
| 50/200 | 0.70 | 0.12 |
| 100/200 | 0.64 | 0.12 |

## C    APPENDIX C

### C.1    EGO STATUS

Unlike methods such as ParaDrive, which directly encode the ego status (can-bus-info) as features, we only encode the predicted ego velocity from the previous frame into our ego features. The ego status is only used as the GT for ego trajectory prediction (as shown in our code). Thus, we added official experimental results of ParaDrive (without ego status encoded and concatenated) in our Table 1. Compared to PareDrive, we achieved significant improvement.

### C.2    IMPLEMENTATION DETAILS

Our baseline utilizes ResNet He et al. (2015) and FPN Lin et al. (2017) as image backbone. We directly encode the 3D features from image features without constructing BEV hidden features, then proceed with our cascaded model. We perform our experiments on 8 A800 GPUs, with a total batch size of 120, aiming to prevent local optima or divergence issues. We also use the AdamW Loshchilov & Hutter (2019) optimizer with an initial learning rate of 5e-4 and a weight decay of 1e-3.

### C.3    EVALUATION MATRIX

For open-loop evaluation, currently, L2 distance and collision rate are adopted to evaluate the smoothness of the predicted trajectory. The L2 distance is used to calculate the deviation between the predicted and ground truth trajectories. The collision rate assesses whether the ego vehicle drives safely within the drivable area, avoiding collisions with surrounding vehicles and road boundaries. In this study, we conduct several studies to comprehensively assess the performance of our model.

For closed-loop evaluation, to ensure fair comparisons with other methods, official CARLA metrics are used. Specifically, Route Completion (RC) evaluates the percentage of the route distance completed by the ego vehicle. Infraction Score (IS) indicates the number of infractions occurring along the route, including collisions with vehicles, pedestrians, road boundaries, etc. Driving Score (DS) is a composite evaluation metric.

In this closed-loop evaluation, we use CARLA to simulate realistic traffic situations with different challenging scenarios, such as obstacle avoidance, complex intersections, and multi-agent interactions. Intermediate results are ignored, and only the final driving outcomes of our method are evaluated.

Table 12: **Open-loop evaluation of planning performance on CARLA dataset.** We evaluate ADDI in complex CARLA scenarios, where it achieves state-of-the-art performance. Note that, the Average L2 distance and Collision Rate are averaged over predictions made within a 2-second horizon at a frequency of 2 Hz. All experiments are reproduced with our collected CARLA dataset.

| Methods | L2 (m) ↓ Avg. | CR (%) ↓ Avg. |
|---|---|---|
| UniADHu et al. (2023) | 0.73 | 0.23 |
| VADJiang et al. (2023) | 0.91 | 0.27 |
| ThinkTwiceJia et al. (2023b) | 0.95 | 0.29 |
| DriveAdapterJia et al. (2023a) | 1.01 | 0.34 |
| SparseDriveSun et al. (2024) | 0.80 | 0.27 |
| Proposed-small | 0.76 | 0.25 |
| Proposed | **0.70** | **0.21** |

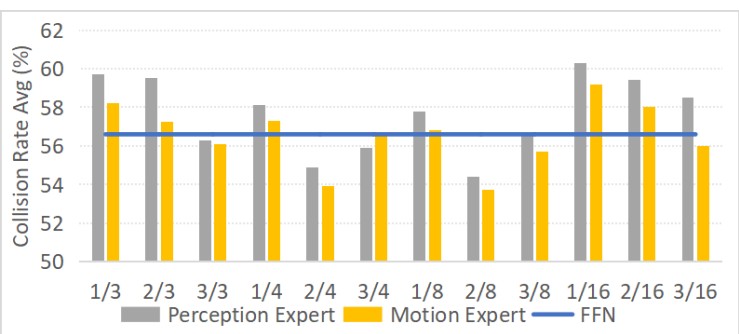

Figure 3: Studies on the perception expert quota (planning use FFN), motion expert quota (perception use FFN) and both FFN.

We use the Bench2Drive benchmark (using leaderboard 2.0). One reason is that when we reproduce other existing methods, we observed that some approaches, such as DriveAdapter and ThinkTwice, use 4 cameras. However, in real AD scenarios like nuScenes, 6 cameras are preferred. Moreover, as discussed in Bench2Drive, existing benchmarks (Town05long and Longest6) evaluate the performance of AD systems by averaging scores across several routes, leading to high variance in the driving score metric. Besides, existing methods are trained and evaluated on their own collected data, making direct comparisons challenging. Bench2Drive provides 44 scenarios by designing 5 distinct short routes (around 150 meters in length) per scenario, each featuring different weathers and towns, which result in a total of 220 routes. Bench2Drive provides a fair and unified benchmark for closed-loop evaluation.

## C.4 CARLA DATASET COLLECTION

The closed-loop CARLA dataset was collected using CARLA Leaderboard v2, which presents significant challenges. CARLA Leaderboard v2 spans 15 towns, encompassing over 90 routes coupled with various scenarios. Each scenario includes several clips comparable to those in nuScenes.

Sensors were configured similarly to those in nuScenes. Six wide-angle cameras were positioned (front-left, front, front-right, rear-left, rear, and rear-right), each with a resolution of 900x1600. Since CARLA v2 cameras employ Brown-Conrady distortion, distortion parameters similar to those of pinhole cameras were simulated. Images were undistorted before being processed by the network. Additionally, a Lidar sensor operating at 10 Hz was employed.

Ego status parameters, including velocity, acceleration, and yaw angle, significantly benefit the planning task. GPS and IMU sensors were used to estimate both the ego and global status. Furthermore, an HD map was simulated for this dataset. The model was trained using the CARLA dataset and subsequently tested on the CARLA simulator under closed-loop evaluation.

## C.5 INTERACTION OPTIMIZATION

Traditional end-to-end autonomous driving methods typically utilize six interactions: Agent-Map, AgentTrajectory-Agent, AgentTrajectory-Map, EgoTrajectory-Agent, EgoTrajectory-Map, AgentTrajectory-EgoTrajectory. These approaches disrupt the cohesion of autonomous driving by decomposing these processes and then linking them through interactions, leading to suboptimal and inefficient practical applications.

In the proposed ADDI, agents and map elements are processed jointly using the Tracking&Perception module, exhibiting implicit interactions facilitated by self-attention mechanisms. Similarly, agent and ego trajectories interact implicitly within the Motion Planning module. Then, we simplify the complex and cumbersome interaction operations by interacting with the outputs (agent and map element features) of Tracking&Perception with the outputs (agent trajectories and ego trajectories) of the Motion Planning module, this simple explicit interaction—agents + map elements vs agent trajectories + ego trajectories—replaces four traditional interactions: AgentTrajectory-Agent, AgentTrajectory-Map, EgoTrajectory-Agent, EgoTrajectory-Map. In addition, a Motion-Temporal explicit interaction is used to aggregate historical motion information, reflecting the inherent temporal interrelations of vehicle movement. This Motion-Temporal interaction also can be used in other methods.

## C.6 LIMITATIONS AND FUTURE WORKS

**Limitations.** The unified module predicts agents and map elements simultaneously, which may result in suboptimal map construction performance. A more effective strategy is required to enhance the accuracy and reliability of online map construction. Second, during module inference, K experts are selected from N, leaving the remaining N-K experts inactive, but still consuming memory resources.

**Future Works.** Future work will focus on further optimizing the inference speed of ADDI. We also plan to incorporate additional static elements, such as traffic lights, lane markings, and traffic signs, to improve the model's ability to handle complex driving scenarios. Expanding the perception range will also be prioritized to accommodate high-speed driving and long-range predictions, ensuring the model's adaptability to diverse environments.

## C.7 THE USE OF LLMS

This project only uses a large language model (LLM) to correct grammatical, morphological, and syntactic errors in Polish text.

## C.8 ETHICS STATEMENT

We followed the ICLR Code of Ethics. Our work uses public datasets with proper attribution. The proposed approach does not involve experiments on human subjects, personal data collection, or sensitive attributes.

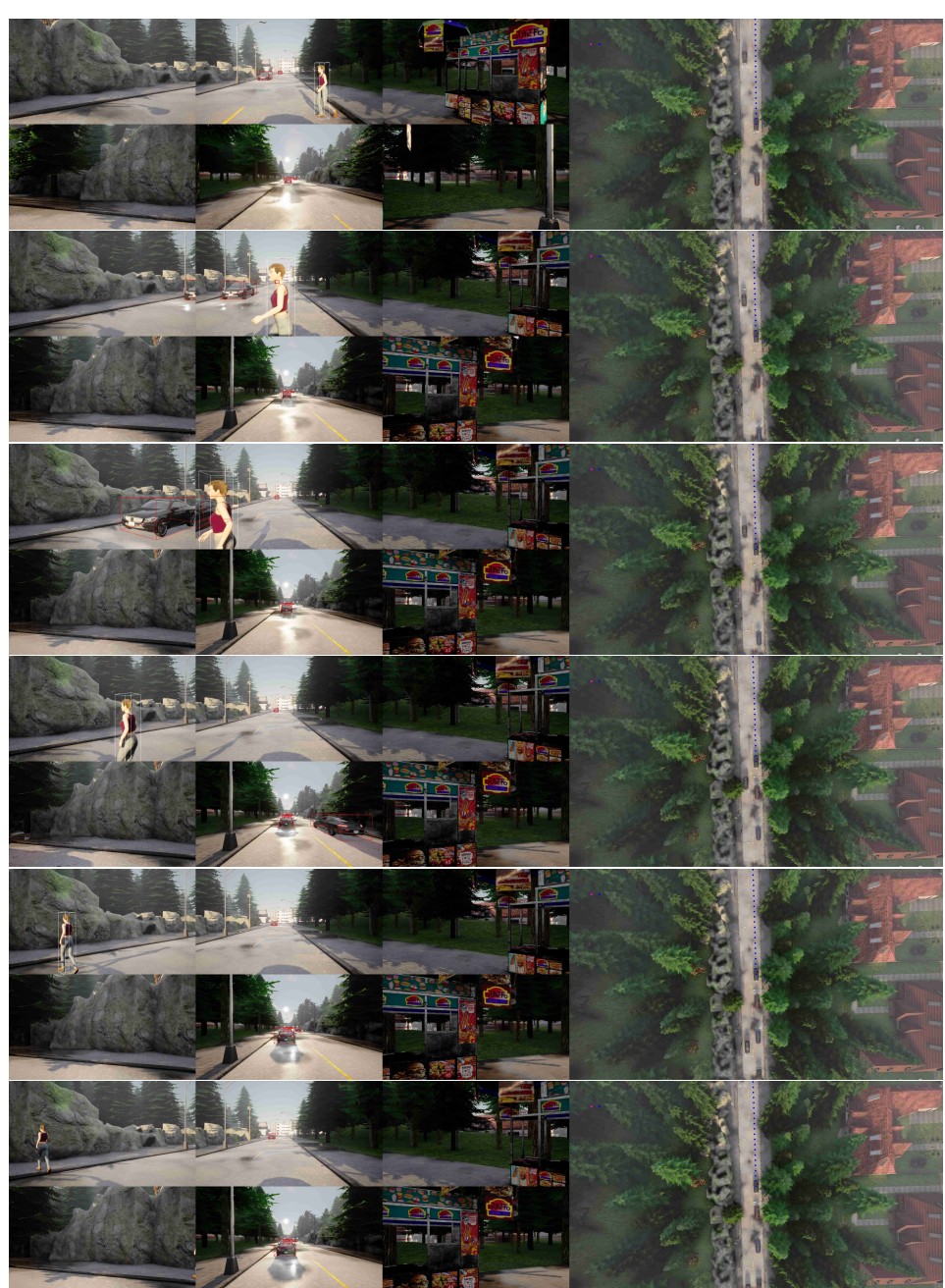

Figure 4: Qualitative results on the CARLA closed-loop experiments-Avoiding pedestrians.

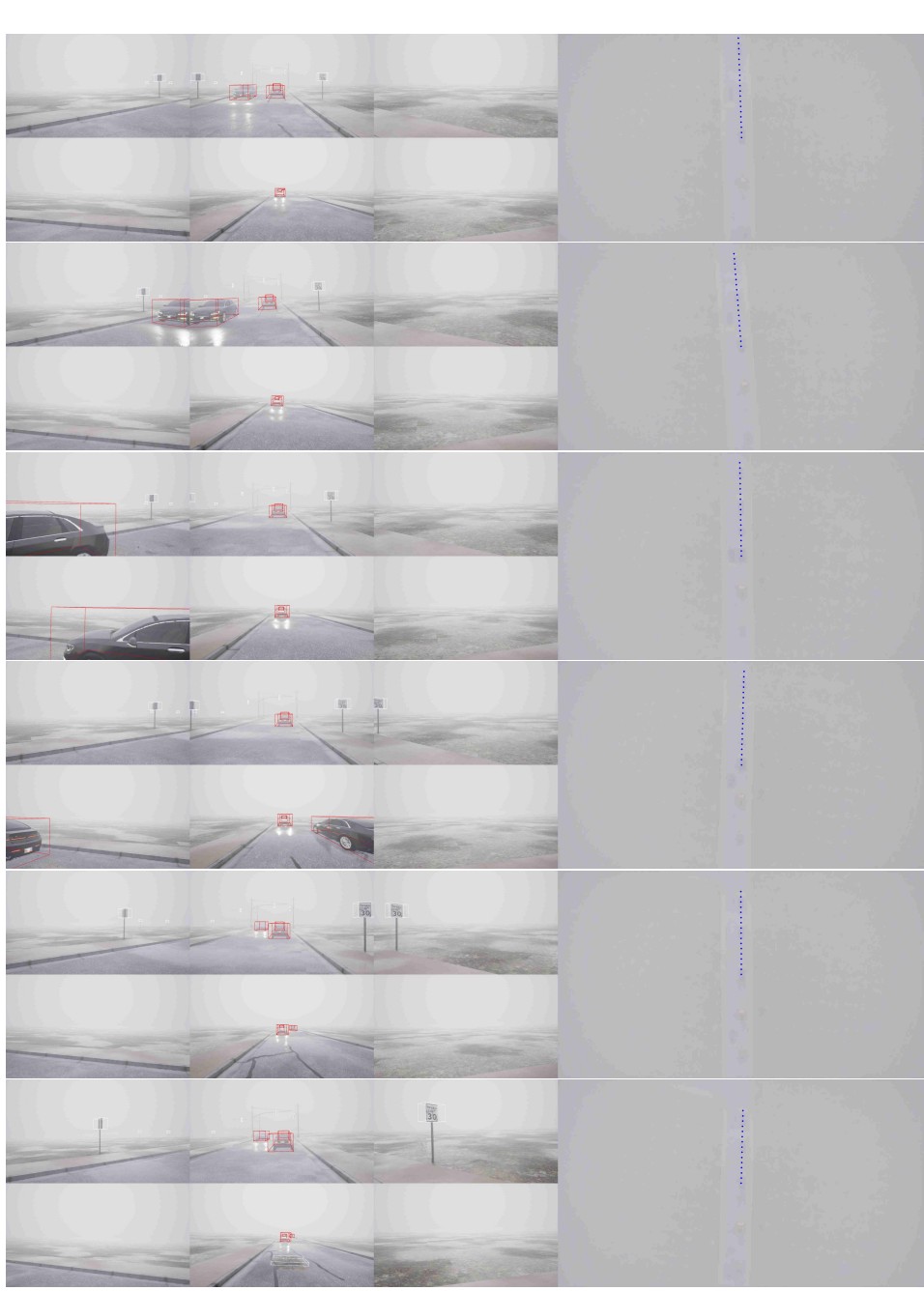

Figure 5: Qualitative results on the CARLA closed-loop experiments-Foggy weather.

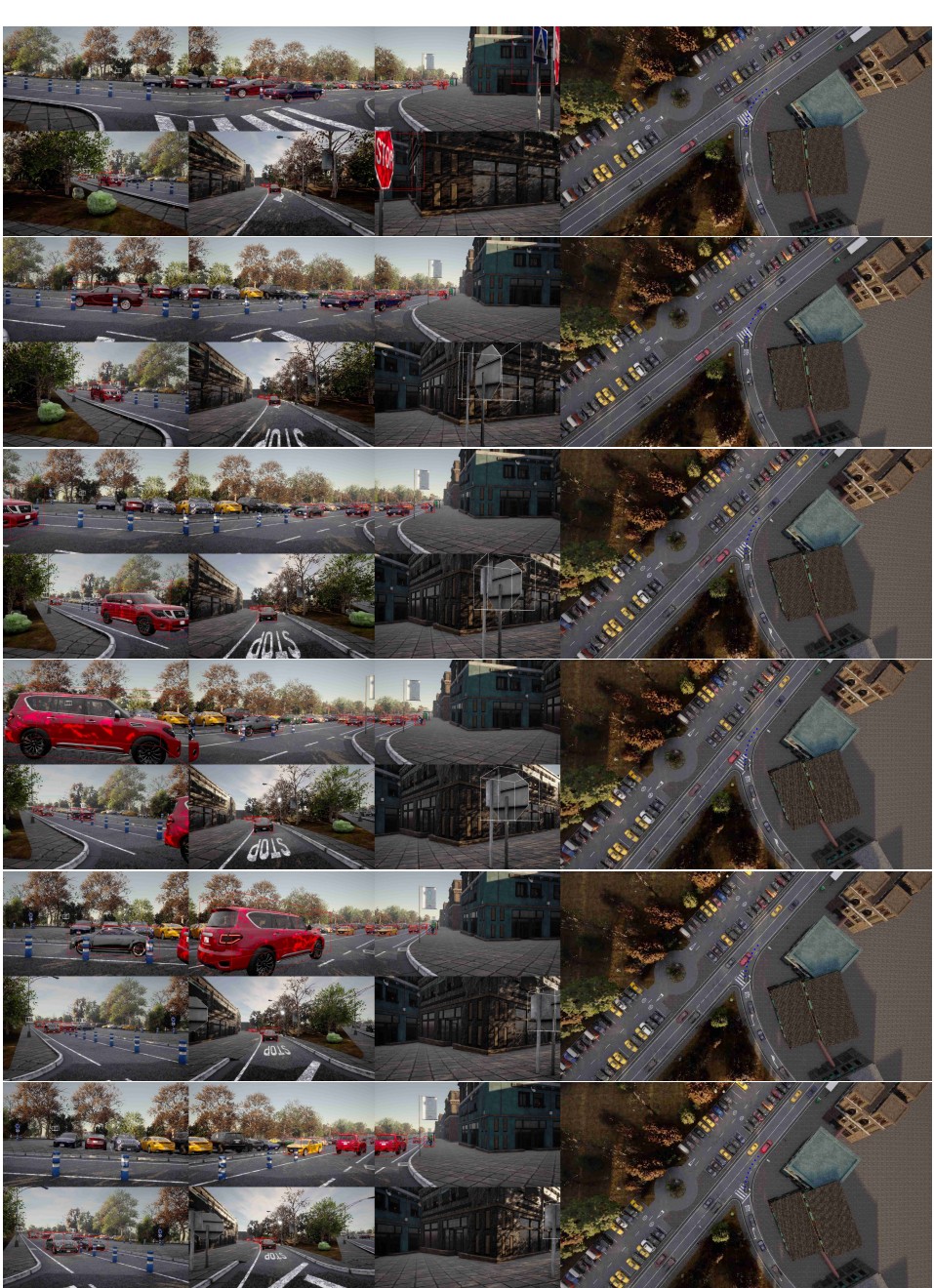

Figure 6: Qualitative results on the CARLA closed-loop experiments-Turning.

