# OpenReview forum: "ADDI: A Simplified E2E Autonomous Driving Model with Distinct Experts and Implicit Interactions"
_ICLR.cc/2026/Conference — ICLR 2026 Conference Withdrawn Submission_

### Official Review · Reviewer_6nSb · 2025-10-21

**Soundness:** 2
**Presentation:** 1
**Contribution:** 2
**Rating:** 2
**Confidence:** 4

**Summary:**

The paper presents ADDI, a end-to-end autonomous driving model that implicits the interactions between four sub-tasks (tracking-by-detection, online mapping, prediction, and planning) with unified modules and distinct experts. ADDI uses a unified detection module for tracking-by-detection and online mapping tasks while uses a motion planning model for agent motion prediction and ego planning tasks. Distinct experts are introduced for each sub-tasks. ADDI achieves state-of-the-art performance on both open-loop and closed-loop benchmarks.

**Strengths:**

1. The model reaches state-of-the-art performance on both open-loop and closed-loop benchmarks.

**Weaknesses:**

1. The details of experts for each task are unclear, including their structures and the differences between experts for different tasks. It is hard to evaluate the expert design without the information.

2. The idea of task unification in autonomous driving is not new. For example, SparseDrive [1] unifies the motion prediction and ego planning in a single module and DriveTransformer [2] conducts all four sub-tasks with a unified transformer decoder parallelly. Thus, the unified model structure of ADDI lacks novelty.

3. The meaning of perception queries is ambiguous. In DETR, each token in perception query represents the box of a single object and tries to capture information of the object in the network. However, in ADDI, a token is input to both detection and online mapping experts. This means it represents an agent and a map element at the same time. The ambiguous representation may make model confused and leads to ineffective optimization.

4. Since ADDI is an end-to-end autonomous driving model, it is better to focus on end-to-end methods in related work, instead of to list works in each sub-tasks.

5. Two lines overlap at L048 and are almost impossible to read.

[1] Sparsedrive: End-to-end autonomous driving via sparse scene representation (ICRA 2025)

[2] Drivetransformer: Unified transformer for scalable end-to-end autonomous driving (ICLR 2025)

**Questions:**

1. The motion query is obtained by concatenating agent trajectory query and ego query, so the task of each token in the query is already distinct (agent query for motion prediction and ego query for planning). Why does ADDI use experts to distinguish between two tasks again?

2. How long is the perception query in ADDI?

---

### Official Review · Reviewer_Auja · 2025-10-29

**Soundness:** 2
**Presentation:** 2
**Contribution:** 2
**Rating:** 2
**Confidence:** 5

**Summary:**

This paper proposes an end-to-end autonomous driving method named ADDI, which simplifies the traditional four subtasks (detection, online mapping, prediction, and planning) into two main modules, i.e. Tracking&Perception and Motion Planning. By introducing two implicit and two explicit interactions, ADDI optimizes the information flow between modules. Experimental results demonstrate that ADDI achieves superior performance and inference efficiency on the nuScenes and CARLA datasets, showing its potential for complex driving scenarios.

**Strengths:**

- Module Integration and Simplification

The paper reduces the complexity of traditional pipelines by unifying the tasks into two modules, improving resource efficiency and system coherence.

- Interaction Optimization

The introduction of implicit (self-attention-based) and explicit interactions (Perception-Motion and Motion-Temporal interactions) enhances efficiency and maintains task consistency.

- Strong Experimental Results

ADDI outperforms existing methods (e.g., SparseDrive, UniAD) across multiple evaluation metrics on both nuScenes and CARLA datasets, while achieving a faster inference speed of 6.2 FPS.

**Weaknesses:**

- Limited Novelty

The idea of multi-task aggregation is not particularly novel. Similar approaches, such as DriveTransformer, also integrate multiple tasks by feeding task-specific queries into a unified decoder for interaction.

- Outdated Baselines

The comparison methods (e.g., UniAD, SparseDrive) are not the most recent or highest-performing algorithms. The authors should include comparisons with more up-to-date methods.

- Insufficient Benchmarks

Experiments are only conducted on the nuScenes and CARLA datasets. The authors should evaluate their method on additional benchmarks, such as NavSim, to demonstrate its generality and robustness.

- Poor Writing Quality

The paper contains noticeable writing issues, such as repeated sentences (e.g., line 48 contains two identical lines) and overly generic phrases, which suggest the use of a large language model. The authors need to carefully refine and polish the writing to improve clarity and professionalism.

**Questions:**

see above weakness

---

### Official Review · Reviewer_jGEK · 2025-10-30

**Soundness:** 2
**Presentation:** 1
**Contribution:** 2
**Rating:** 2
**Confidence:** 4

**Summary:**

This paper presents ADDI, an end-to-end autonomous driving framework aimed at addressing the inefficiencies of traditional modular pipelines. The authors argue that conventional systems, which separate tasks like tracking, mapping, prediction, and planning into distinct modules, suffer from suboptimality due to the complex interactions required to link them. To mitigate this, the proposed ADDI method employs a unified architecture. Specifically, it integrates tracking-by-detection and online mapping into a single detection module that uses expert designs to output both results simultaneously. In a similar fashion, it unifies agent trajectory prediction and ego-vehicle planning within a single motion planning model. The authors claim that this unified structure eliminates the need for most inter-module interactions, relying instead on a simplified set of two implicit and two explicit mechanisms.

**Strengths:**

1. The proposed paradigm consolidates multiple driving-related tasks into a streamlined two-module architecture, comprising a unified 'Tracking & Perception' module and a 'Motion Planning' module. By leveraging specialized expert designs within these integrated models, this approach effectively enhances overall performance.

2. The proposed method achieves a more robust representation of temporal features by redesigning how modules communicate. Instead of relying on complex and cumbersome traditional interactions, a simplified framework is introduced built on two implicit and two explicit interactions.

**Weaknesses:**

1. The paper's writing has a large room for improvement, including messed-up formatting at line 048, wrong citation format, wrong formula format (italic letter should be used for variables), and inconsistent highlighting in the experiment section (in Table 1 and Table 3, some columns have bold while others don't).

2. Limited Novelty. Many contents in the Method section are adapted from previous works. Temporal aggregation is already well studied in other sparse detection works, including StreamPETR[1], Sparse4d[2], SparseDrive[3]. Spatial aggregation with implicit interaction of agent tokens, mapping tokens, and ego tokens is shown in DriveTransformer[4].

3. The motivation behind the Mixture-of-Expert design is unclear. The paper says in line 267 that "Since we use a unified model to predict both agents and map elements simultaneously, using an identical approach to fit both agents and map elements with significantly different structural dimensions, is inherently imprecise." The unified model is only for information exchange; there are parameters dedicated to each modality ,such as the supervision head after the transformer layer. Also, since the different tokens are heterogeneous (in that they are supervised and initialized differently with different embeddings), why use a learnt router to direct different tokens to different mlps instead of directly assigning different tokens to different mlps.

[1] Exploring Object-Centric Temporal Modeling for Efficient Multi-View 3D Object Detection, Wang, et.al.
[2] Sparse4D: Multi-view 3D Object Detection with Sparse Spatial-Temporal Fusion, Lin, et.al.
[3] SparseDrive: End-to-End Autonomous Driving via Sparse Scene Representation, Sun, et. al.
[4] DriveTransformer: Unified Transformer for Scalable End-to-End Autonomous Driving, Jia, et. al.

**Questions:**

1. The best part of MoE is that scaling up expert numbers will boost performance while maintaining the same efficiency. It would be great to see such experiments.

2. In the ablation study, without P-M interaction, the motion tokens can't see perception tokens, which is analogous to driving blindly; how can this setup still achieve 40 driving scores?

---

### Official Review · Reviewer_s69D · 2025-10-30

**Soundness:** 2
**Presentation:** 2
**Contribution:** 2
**Rating:** 2
**Confidence:** 4

**Summary:**

This paper introduces ADDI, which unifies multiple tasks with Tracking&Perception and Motion Planning. Experiments on the open-loop and closed-loop evaluations demonstrate that this method achieves state-of-the-art performance.

**Strengths:**

ADDI simplifies the traditional four tasks (tracking-by-detection, online mapping, prediction, and planning) into two stages (Tracking&Perception and Motion Planning). ADDI also utilizes two implicit interactions and two explicit interactions, reducing the complexity of existing interactions.

**Weaknesses:**

1.  The format on line 48 is incorrect.
2.  Lack of related work discussion about MoE in AD.
3.  In the Perception Expert, how do you distinguish between the agent expert and the map expert? How are their numbers determined, and how is load balancing handled?
4.  In Motion Planning, the same questions as in (2) apply—how are experts differentiated, how many are used, and how is load balancing achieved?
5.  Ablation studies on the number of experts and the number of layers.
6.  What is the architecture of the Implicit Interaction? How does it differ from self-attention?
7.  The paper lacks visualization of other agents’ trajectory predictions.

**Questions:**

same to Weakness

---

### Note · Authors · 2025-11-12

I have read and agree with the venue's withdrawal policy on behalf of myself and my co-authors.